# SPHK1/S1PR1/PPAR-α axis restores TJs between uroepithelium providing new ideas for IC/BPS treatment

Junjie Zhang[1,2,*] ⓘ, Qingyu Ge[1,2,*], Tianpeng Du[1,2,*], Yuhao Kuang[1,2], Zongyao Fan[1,2], Xinyi Jia[3], Wenjin Gu[4], Zhengsen Chen[1,2], Zhongqing Wei[1,2] ⓘ, Baixin Shen[1,2] ⓘ

Interstitial cystitis/bladder pain syndrome (IC/BPS) represents a chronic, aseptic inflammatory bladder condition with an unclear etiology and few therapeutic options. A composite barrier structure composed of the uroepithelium and glycosaminoglycan layer forms on the bladder's inner surface to block urine and other harmful substances. Dysfunction of this barrier may initiate the pathogenesis of IC/BPS. Sphingosine-1-phosphate (S1P) plays a crucial role in forming tight junctions. Perfusion of S1P into the bladder restored uroepithelial tight junctions in mice with cyclophosphamide-induced acute cystitis and ameliorated symptoms of the lower urinary tract. Mice lacking sphingosine kinase 1 (SHPK1) exhibited more severe bladder injuries and dysfunction. Concurrent in vitro experiments elucidated S1P's protective effects and its role as a primary messenger through SPHK1 and S1P receptor 1 (S1PR1) knockdown. This study identifies a novel mechanism whereby S1P binding to S1PR1 activates the PPAR-α pathway, thereby enhancing cholesterol transport and restoring tight junctions between uroepithelial cells. These findings elucidate the regulatory role of S1P in the bladder epithelial barrier and highlight a promising therapeutic target for IC/BPS.

## Introduction

Interstitial cystitis/bladder pain syndrome (IC/BPS) is a chronic aseptic cystitis with unknown etiology and poorly understood pathological mechanisms (1). Patients with IC/BPS experience extreme urinary frequency, urgency, and severe pain during bladder filling, which leads to considerable suffering. In severe cases, the surgical removal of the bladder may be considered, significantly affecting patients' quality of life (2, 3). Despite the availability of various treatments such as bladder instillations and oral supplementation with glycosaminoglycan (GAG) analogs, their efficacy remains below expectations (4, 5). Consequently, identifying new therapeutic targets for IC/BPS remains a critical need.

The bladder epithelial barrier, a composite structure comprising a GAG layer and a uroepithelial cell layer, segregates urine from surrounding tissues (6, 7). The GAG layer, containing hydrophobic molecules such as hyaluronate, chondroitin sulfate, and dermatan sulfate, forms a protective membrane on the bladder surface (6, 8). Adjacent to the urine, the umbrella cell layer forms tight junctions, enhancing cellular cohesion. The umbrella cells' tips contain distinctive lipids and vacuolar plaque proteins, composing a parietal membrane that blocks harmful substances and pathogens from penetrating subepithelial tissues (7, 9, 10). Intermediate and basal uroepithelial cells proliferate and differentiate, thereby replenishing the umbrella cell layer (11). Disruption of the vascular endothelial barrier results in leakage and various cranial and pulmonary disorders. Similarly, damage to the bladder epithelial barrier causes urine infiltration and subsequent pathologies (12, 13, 14). Elevated bladder epithelial permeability, a key pathophysiological change in interstitial cystitis, leads to lower urinary tract symptoms (LUTS) (15). Current treatments primarily focus on symptomatic relief through GAG analog supplementation, often neglecting the repair of epithelial intercellular junctions. Targeting the repair of epithelial cell junctions represents a novel therapeutic approach for IC/BPS treatment.

Our research demonstrated a significant enrichment of sphingosine-1-phosphate (S1P) in the urine of IC/BPS patients, corroborated by Asi et al who reported similarly elevated serum levels (16, 17). These findings indicate a potential role for S1P in the pathophysiology of IC/BPS, though its precise function is yet to be elucidated. S1P, an active sphingolipid metabolite, is essential for cell migration, adhesion, survival, and proliferation. S1P is synthesized intracellularly from sphingomyelin by sphingosine kinases SPHK1 and SPHK2 and degraded by S1P lyase (SPL) and S1P phosphatases (SPP) to maintain stable intracellular and extracellular concentrations (18). S1P activates the S1P receptor 1-5 (S1PR1-5) on cell membranes via autocrine/paracrine action, acting

[1]Department of Urology, The Second Affiliated Hospital of Nanjing Medical University, Nanjing, China   [2]Department of Urology, The Second Clinical Medical College of Nanjing Medical University, Nanjing, China   [3]Respiratory Department, Children's Hospital of Nanjing Medical University, Nanjing, China   [4]Nanjing Medical University, Nanjing, China

Correspondence: Weizq11@163.com; baixinshen@njmu.edu.cn
*Junjie Zhang, Qingyu Ge, and Tianpeng Du contributed equally to this work

as a first messenger and significantly influencing lymphatic circulation, cardiovascular development, and intercellular tight junction maintenance (19, 20). In addition, S1P influences an intracellular second messenger system essential for cell motility, proliferation, and anti-apoptosis (21). The extracellular concentration of S1P is regulated by the combined transport mechanisms of Mfsd2a and Spns2 (22). Low extracellular S1P concentrations result in cerebrovascular leakage; however, supplementation with exogenous S1P restores vascular endothelial barrier function, highlighting its protective effects (22, 23, 24). The barrier-protective effects of exogenously injected S1P in brain tissues diminish with S1PR1 knockdown, indicating primary mediation through S1PR1 activation (22, 24). After bleomycin-induced lung injury in mice, S1PR1 knockdown resulted in exacerbated pulmonary vascular endothelial barrier disruption, along with increased inflammation and collagen deposition (25). Consequently, S1P primarily maintains barrier integrity through S1PR1 activation, and S1PR1 knockdown disrupts intercellular tight junctions, increasing cell permeability.

In this study, we found that exogenous S1P effectively restored the integrity of the urinary epithelium barrier disrupted by inflammatory conditions. However, this restorative effect was nullified when S1PR1 was knocked down. Upon activation by S1P, S1PR1 engages and activates the PPAR-α pathway, which enhances cholesterol transport through lipoproteins and thus reinstates the tight junctions among urothelial cells. This discovery reveals a novel mechanism for the restoration of epithelial barriers.

# Results

## Bladder perfusion with S1P improves LUTS of cyclophosphamide (CYP)-induced cystitis in mice

A single intraperitoneal injection of 150 mg/kg CYP was administered, followed 24 h later by bladder perfusion with either S1P or a vehicle control. Thirty minutes post-perfusion, the bladders were manually emptied, and behavioral experiments or bladder harvesting followed 4 h later (Fig 1A). S1P was perfused at various concentrations: 0, 50, 100, 200, and 400 µM, creating a concentration gradient. We found that urination in mice returned to near-normal levels at a 400 µM concentration of S1P in bladder perfusion (Fig 1B). The 400 µM perfusion concentration was chosen for subsequent experiments. Congestion and edema typically characterize CYP-induced acute cystitis. The bladder weight ratio serves as an indicator of bladder edema severity. After S1P perfusion, the gross appearance of the bladder improved, and the bladder weight ratio decreased (Fig 1C and D). Subsequent pathological analysis involved sections and HE staining of mouse bladders. In the CYP group, uroepithelial exfoliation and significant subepithelial tissue edema were observed, whereas in the S1P-perfused group, uroepithelial proliferation and repair were noted (Fig 1E).

Voiding spot assays (VSA) recorded the frequency and volume of urination in mice. Urine volume was quantitatively analyzed using an area-volume standard curve derived from the same batch of filter paper (Fig S1A). Mice in the CYP group exhibited increased urination frequency and decreased average urine volume,

mirroring symptoms of frequent urination in IC/BPS. These conditions improved after bladder perfusion with S1P (Fig 1F–I). The von Frey filament score, indicative of mice's lower abdominal sensitivity, corresponds to pelvic pain symptoms in IC/BPS. This sensitivity was alleviated after S1P perfusion, which countered the increased lower abdominal sensitivity induced by CYP (Fig 1J and K). To minimize environmental and consciousness effects on urination, urodynamic tests were performed under 2% isoflurane anesthesia (Fig 1J–O). The results indicated a shortened storage period and weakened maximal forced urethral muscle contraction in the CYP group, both of which were restored in the S1P-treated group (Fig 1J–O). These findings demonstrate that bladder perfusion with S1P can mitigate CYP-induced acute cystitis in mice.

## The urothelium and interepithelial tight junctions were restored after S1P treatment in mice

We next examined the effects of S1P on the uroepithelial layer. To isolate the uroepithelial layer from the effects of bladder smooth muscle proteins, we manually separated this layer and extracted its proteins under a dissecting microscope. Western blot analysis revealed that S1P perfusion restored interepithelial tight junctions (TJs) in the uroepithelial layer, including ZO-1, occludin, and claudin-4. In addition, expression of the epithelial marker E-cadherin was increased (Fig 2A and B). We further analyzed the distribution of TJs by quantifying and localizing ZO-1, occludin, and claudin-4 through immunohistochemistry (IHC). The results indicated significant thinning and reduced expression of TJs in the bladders of the CYP group. In contrast, S1P perfusion enhanced both the expression and distribution of these junctions (Fig 2C). IHC for Krt20, a uroepithelial cell marker, revealed a reduction in uroepithelial cell layers in the CYP group. Conversely, this damage was repaired after S1P perfusion (Fig 2D).

The uroepithelium is composed of three layers: a basal cell layer, an intermediate cell layer, and an umbrella cell layer. The umbrella cell layer, being closest to the urine, serves as a crucial barrier that isolates the urine. Basal cells replenish the umbrella cell layer by differentiating, thereby serving as a cellular reserve. These layers are critical indicators for assessing epithelial function (9). We conducted multiplex IF assays using UPK IIIa (umbrella cell marker) and Krt5 (basal cell marker). In the CYP-treated mice, the umbrella cell layer was absent, and the basal cell layer was diminished, indicating significant epithelial damage. Conversely, in the S1P-perfused group, both the umbrella and basal cell layers showed significant restoration (Fig 2E). By verifying the protein expression of S1PR1-5 in the bladder of mice in the S1P bladder perfusion group, we found that S1PR1 expression was elevated and S1PR2-5 was not significantly altered (Fig S1B). Then, analysis of inflammatory indices revealed that S1P perfusion decreased inflammation and reduced apoptosis in bladder tissues (Fig S1C–E). IHC for Ki-67 indicated enhanced cellular proliferation in the S1P-treated group (Fig S1F). These findings demonstrate that S1P bladder perfusion promotes restoration of the uroepithelial layers and intercellular tight junctions, while also alleviating inflammation and apoptosis in bladder tissues. However, the independent restoration of intercellular tight junctions could not be assessed in our

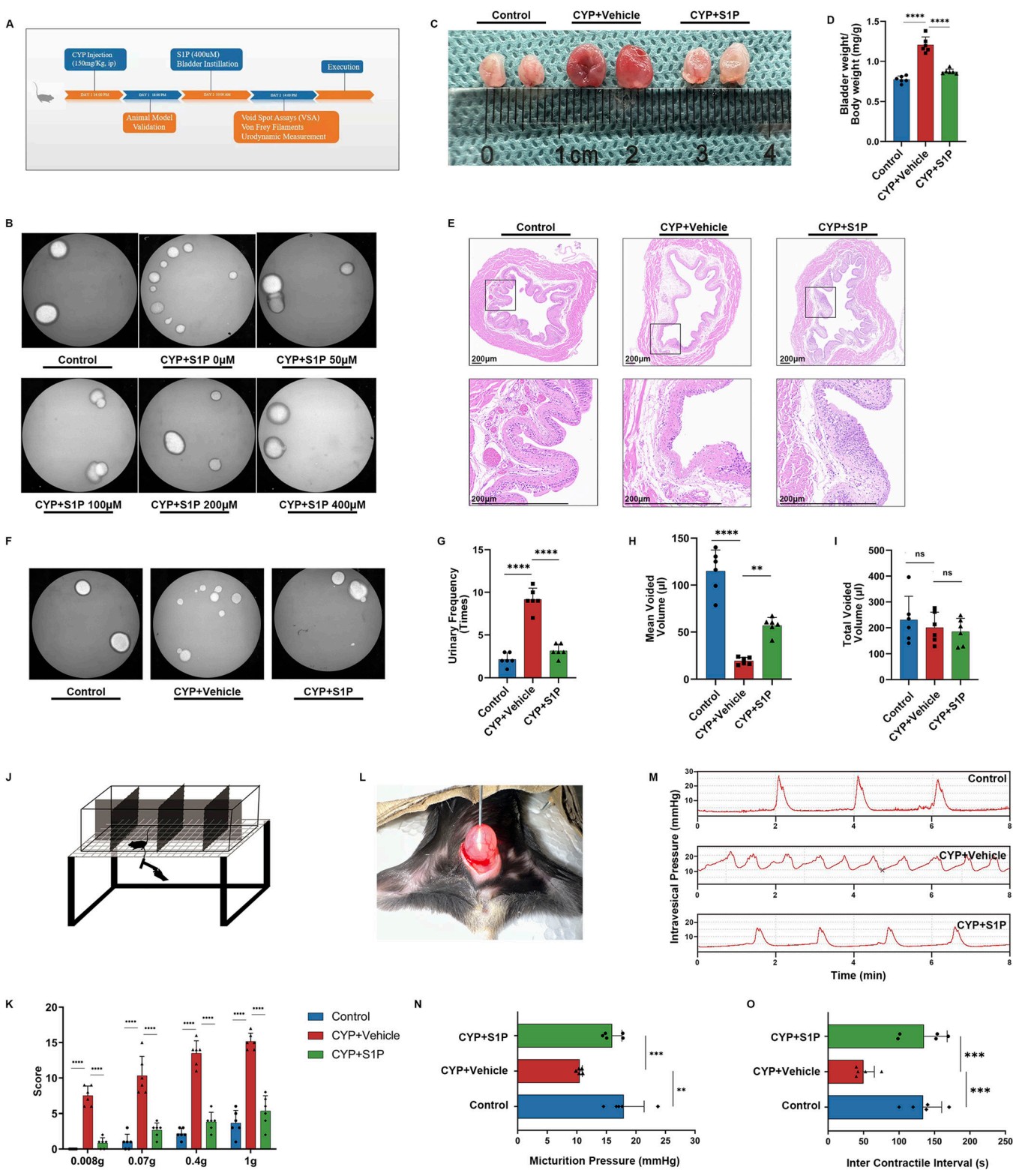

**Figure 1. S1P perfusion ameliorates the manifestation of cyclophosphamide-induced acute cystitis in mice.**
**(A)** Experimental flowchart of this study. **(B)** Bladder perfusion with S1P concentration gradient explored. **(C)** Comparison of the gross appearance of the bladders.
**(D)** Ratio of the bladders' wet weight to the mice's body weight (n = 6). **(E)** HE staining of the bladder tissues. **(F)** Voiding imprints of the mice, each bright white spot representing a single voiding. **(G)** Voiding imprints were experimented on in the statistical analysis of the number of micturition. **(H)** Voiding imprints were experimented on in the statistical analysis of the average micturition volume. **(I)** Voiding imprints were experimented on in the statistical analysis of the total micturition volume (n = 6). **(J)** Pattern diagram of pain filament experiment. **(K)** Pain score of lower abdominal stimulation in mice (n = 8). **(L)** Pattern diagram of urodynamics in mice.

in vivo experiments; further refinement will be pursued in our in vitro studies.

## SPHK1−/− mice show more severe bladder injury under CYP-induced injury

We used SPHK1−/− mice to establish a model of low S1P levels in vivo. Serum S1P levels were significantly reduced in SHPK1+/− and SHPK1−/− mice (Fig 3A). Bladder tissue S1P levels were elevated in CYP-induced cystitis mice, whereas knockdown of SPHK1 reduced S1P levels (Fig 3B). SPHK1−/− mice exhibited more severe CYP-induced injuries, characterized by increased voiding frequency, reduced mean urine output, and heightened pelvic pain sensitivity (Fig 3C and D). Urodynamic assessments under anesthesia revealed increased mechanical sensitivity to bladder dilatation in SPHK1−/− mice, with a significantly shortened voiding interval compared with the WT CYP group (Fig 3E). Bladder tissues harvested from SPHK1−/− mice were stained with HE to demonstrate severe disruption of the bladder epithelial layer and enhanced subepithelial edema (Fig 3F). Collectively, these behavioral and pathological findings indicate that SPHK1−/− mice experience significantly more severe CYP-induced bladder damage.

After this, our analyses focused on both the tight junctions and the uroepithelium. Tight junction proteins were notably reduced in the SPHK1−/− CYP group compared with the WT CYP group, with diminished distribution and numerous disruptions within the uroepithelium (Fig 3G and H). Concurrently, we observed a decrease in uroepithelial cell number and compromised coverage of subepithelial tissues (Fig 3I). Moreover, the SPHK1−/− CYP group showed an absence of the umbrella cell layer and significant thinning of the basal cell layer, indicative of a loss in the epithelial layer's protective and repair capabilities (Fig 3J). This confirms that S1P exerts a protective effect on uroepithelial tight junctions, effective across varying S1P levels. However, the limitations of our in vivo experiments precluded an independent analysis of uroepithelial intercellular tight junction recovery; we intend to address these limitations in our forthcoming in vitro studies.

## Exogenous addition of S1P protects against LPS-induced destruction of cellular TJs

To model disruption of uroepithelial cell TJs, we continuously stimulated Sv-huc-1 cells with LPS in the culture medium for 24 h. After this, a concentration gradient of LPS (0–4 µg/ml) was applied, and Western blot analyses were performed on barrier-associated TJs (ZO-1, occludin, claudin-4), leading to the generation of dose-response curves. It was observed that TJs expression was most significantly reduced at an LPS concentration of 1 µg/ml (Fig 4A and B). Subsequently, a similar concentration gradient approach was used for S1P (0–400 nmol/liter), where Western blot analyses on TJs also yielded dose-response curves. The most pronounced protective effect on TJs occurred at an S1P concentration of 100 nmol/liter (Fig 4C and D). These concentrations, 1 µg/ml LPS and 100

nmol/liter S1P, were selected for subsequent experiments. Using Western blotting, TJs were quantified, and their distribution was assessed via immunofluorescence. After 24 h of LPS stimulation, a notable decrease and discontinuity in TJs expression were observed, which S1P addition effectively restored (Fig 4E–G). In the epithelial permeability assay, increased FITC leakage was seen in the LPS-treated group but was mitigated by S1P treatment, effectively restoring uroepithelial barrier function (Fig 4H). Collectively, these results demonstrate that exogenous S1P effectively counters LPS-induced disruptions in TJs and enhances intercellular barrier function.

To further validate the role of S1P, we conducted a knockdown of SPHK1, the enzyme responsible for endogenous S1P production. SPHK1-siRNA-1 was chosen after confirming its efficiency in knockdown experiments (Fig 4I). After SPHK1 knockdown and LPS treatment, there was a more pronounced disruption of TJs, evidenced by reduced expression and altered periplasmic distribution compared with the LPS-only group. This disruption was mitigated by subsequent S1P addition, which restored TJs expression and periplasmic structure (Fig 4J–L). In the group treated with LPS and SPHK1 knockdown, FITC leakage was more severe, but this was ameliorated by the addition of exogenous S1P (Fig 4M). The results demonstrated that reduced S1P levels led to more severe TJs disruption under inflammatory conditions. These experiments confirm S1P's protective effect on TJs. However, because S1P operates via both first- and second-messenger pathways, further exploration is needed to delineate which pathway predominates.

## S1P exerts a protective effect on TJs through activation of S1PR1

We concurrently assessed S1PR1 expression after the exogenous addition of S1P and knockdown of SPHK1. It was found that S1PR1 expression increased with the exogenous addition of S1P and decreased after SPHK1 knockdown, suggesting S1P's role in activating S1PR1 (Fig 5A and B). Following co-localization studies by using a fluorescent BODIPY-conjugated S1P probe and S1PR1 indicated that exogenous S1P primarily acts as a first messenger, binding to S1PR1 without nuclear entry to exert its biological effects (Fig 5C). To elucidate S1PR1's role, we conducted a knockdown using S1PR1-si-2, selected based on knockdown efficiency verification (Fig 5D). Upon analyzing the expression and distribution of TJs, we noted a loss of function after S1PR1 knockdown (Fig 5E–G). Furthermore, increased FITC leakage was observed in the epithelial permeability assay after S1PR1 knockdown (Fig 5H). These results demonstrate that S1PR1 is crucial for S1P's protective effects on TJs.

To elucidate the downstream mechanisms after S1PR1 activation, we conducted whole-gene transcriptomics sequencing on cells from both the LPS group and the LPS+S1P group. Using differential gene expression analysis, we identified significant up-regulation of genes associated with the PPAR signaling pathway, including *APOA1*, *APOA2*, and *APOC3* (Fig 5I). GO enrichment analysis revealed active lipid metabolism in the S1P group (Fig 5J), and KEGG enrichment analysis showed activation of cholesterol metabolism

---

**(M)** Urodynamics manometry diagram of mice, with each peak representing a single micturition. **(N, O)** Statistical analysis of the maximal contraction force and micturition intervals in the urodynamics experiment in mice (n = 5). Bars represent mean ± SE, **$P < 0.01$, ***$P < 0.001$, ****$P < 0.0001$ determined by one-way ANOVA.

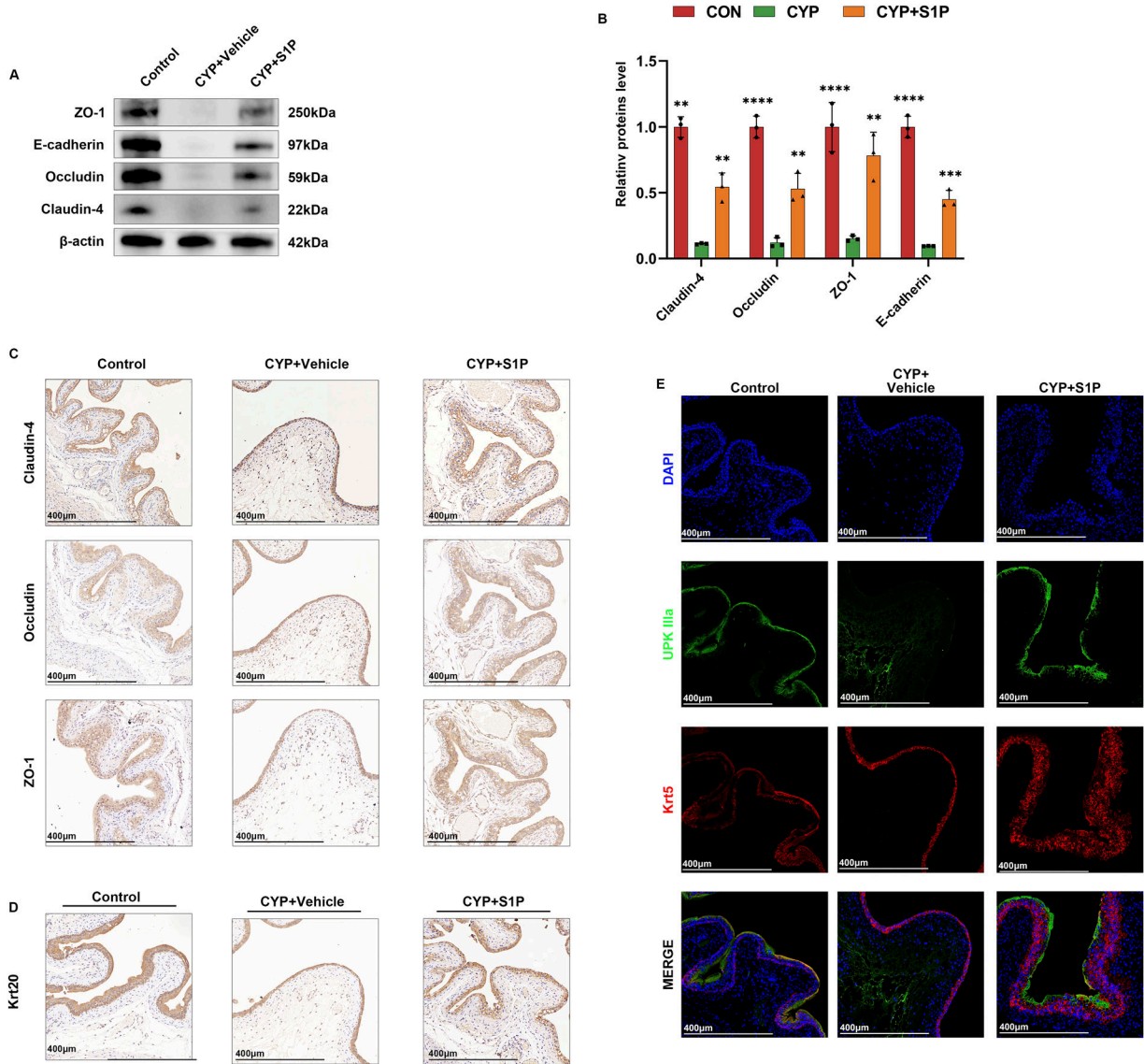

**Figure 2. S1P perfusion restores the epithelial cell layer and intercellular tight junctions in mouse bladder urothelium.**
**(A)** WB analysis of the expression of three TJs: ZO-1, occludin, claudin-4, and the epithelial indicator E-cadherin. **(B)** Semi-quantitative and statistical analysis of the results of A using ImageJ (n = 3). **(C)** Immunohistochemical staining of bladder tissue for ZO-1, occludin, and claudin-4. **(D)** Immunohistochemistry of Krt20 in bladder tissue. **(E)** Multiplex IF of UPK IIIa and Krt5 in bladder tissue. Bars represent mean ± SE, **$P$ < 0.01, ***$P$ < 0.001, ****$P$ < 0.0001 determined by one-way ANOVA. Source data are available for this figure.

and the PPAR signaling pathway (Fig 5K). Considering that *APOA1*, *APOA2*, and *APOC3* are known downstream targets of PPAR-α, we assessed their expression levels as indicators to confirm the activation of the PPAR-α pathway by S1P (26).

### S1PR1 activates the PPAR-α pathway to promote cholesterol transport to restore TJs destruction

We established four experimental groups: with and without LPS, S1P, and S1PR1-siRNA, to investigate the effects of S1P and S1PR1 on PPAR-α and its downstream genes. Western blot analysis revealed that exogenous S1P activated PPAR-α, elevating the expression of lipoproteins APOA1, APOA2, and APOC3. This

activation was abolished after S1PR1 knockdown (Fig 6A), indicating that S1P activates the PPAR-α pathway via S1PR1. To determine whether PPAR-α plays a crucial role in S1PR1's protective effects on TJs, we used PPAR-α-si-3 to knock down PPAR-α (Fig 6B). Upon assessing TJs, we observed that the restoration of TJ expression and distribution by exogenous S1P was abolished after PPAR-α knockdown, indicating that PPAR-α is the principal effector protein mediating biological functions post-S1PR1 activation (Fig 6C–E). Simultaneously, we evaluated the expression of S1PR1 and the downstream lipoproteins of PPAR-α. We found the expression of S1PR1 remained largely unchanged after PPAR-α knockdown; however, the levels of downstream lipoproteins APOA1, APOA2, and APOC3 significantly decreased (Fig 6F).

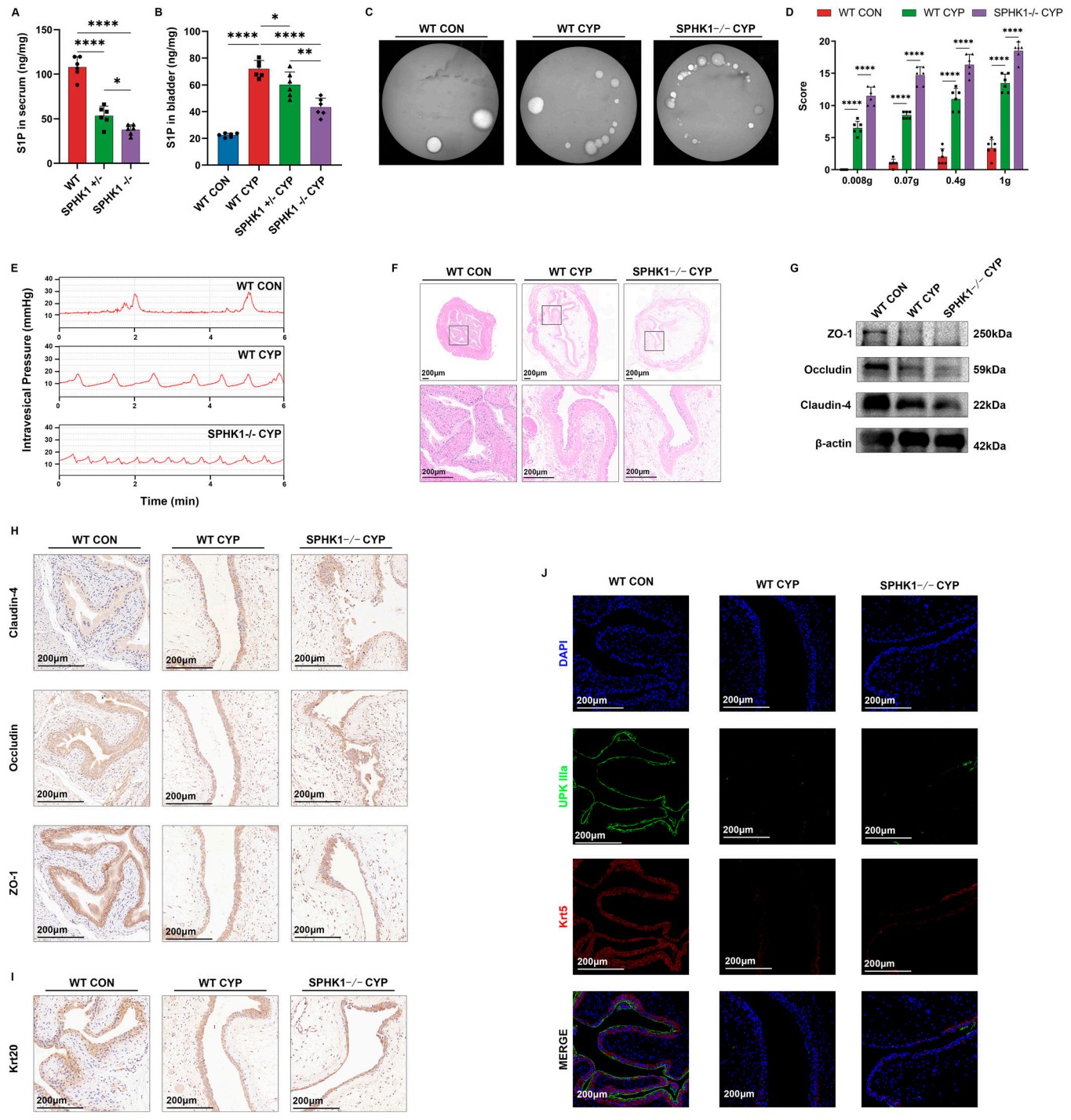

**Figure 3. SPHK1−/− mice bladders underwent more severe damage induced by cyclophosphamide.**
**(A, B)** ELISA assay for S1P in mouse serum and bladder. **(C)** Voiding imprints of the mice, each bright white spot representing a single voiding. **(D)** Mouse lower abdominal stimulation pain scores (n = 6). **(E)** Mouse urodynamic manometry, each peak represents one voiding. **(F)** HE staining of bladder tissues. **(G)** WB analysis of three TJs: ZO-1, occludin, claudin-4. **(H)** Immunohistochemistry staining of bladder tissue for ZO-1, occludin, claudin-4. **(I)** Immunohistochemistry of Krt20 in bladder tissue. **(J)** Multiplex IF of UPK IIIa and Krt5 in bladder tissue. Bars represent mean ± SE, ****$P$ < 0.0001 determined by one-way ANOVA.
Source data are available for this figure.

Furthermore, immunoprecipitation experiments revealed that APOA1, APOA2, and APOC3 interact with each other to facilitate transport functions (Fig 6G).

Cholesterol, a major component of cell membranes, plays an essential role in the distribution and formation of TJs (27, 28). We used Filipin staining to visualize localized cholesterol in cell

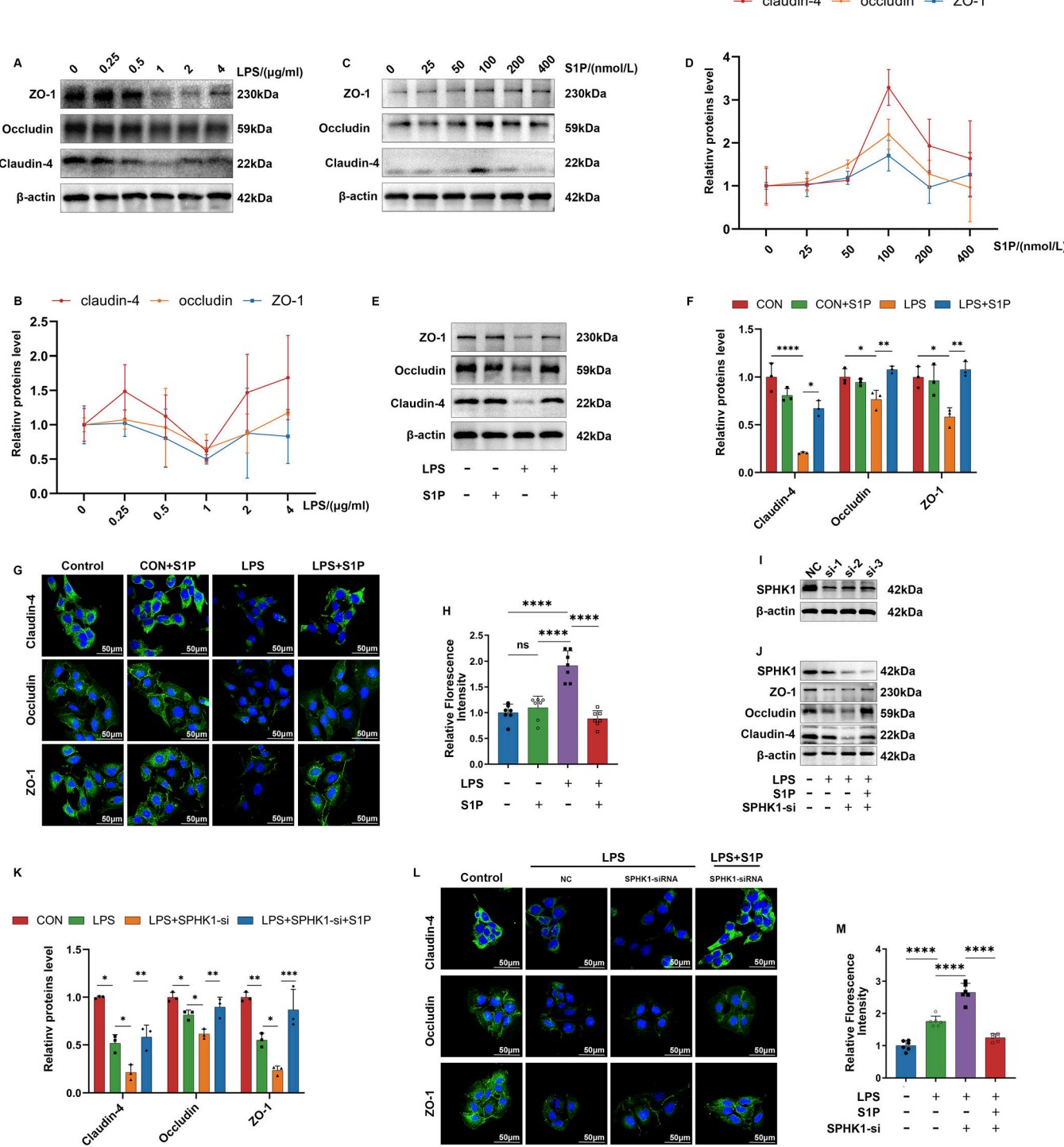

**Figure 4. Exogenous addition of S1P protects against LPS-induced TJs destruction in Sv-huc-1.**
**(A)** Setting LPS concentration gradient 0, 0.25, 0.5, 1, 2, 4 μg/ml to construct a cell model, and WB analysis of TJs was performed. **(B)** Dose effect curve of LPS. (n = 3). **(C)** Setting S1P concentration gradient 0, 25, 50, 100, 200, 400 nmol/liter for Sv-huc-1 under LPS 1 μg/ml intervention and WB analysis of TJs. **(D)** Dose effect curve of S1P. **(E)** WB analysis of TJs under LPS, S1P addition. **(F)** Using ImageJ software to analyze the TJs in E Semi-quantitative analysis of TJs (n = 3). **(G)** IF imaging of TJs under LPS, S1P addition. **(H)** Relative fluorescence intensity analysis of epithelial permeability under LPS, S1P interventions (n = 6). **(I)** Verification of the knockdown efficiency of SPHK1-siRNA. **(J)** After the knockdown of SPHK1 was completed, the three types of cellular proteins in cells with LPS or S1P addition were analyzed by WB. TJs were analyzed by WB. **(J, K)** Semi-quantitative analysis of TJs in (J) using ImageJ software (n = 3). **(L)** IF imaging of TJs under the addition of LPS, S1P after completion of knockdown of SPHK1. **(M)** Relative fluorescence intensity analysis of epithelial permeabilization under the intervention of LPS, S1P after completion of knockdown of SPHK1 (n = 6). Bars represent mean ± SE, *P < 0.05, **P < 0.01, ***P < 0.001, ****P < 0.0001 determined by one-way ANOVA.
Source data are available for this figure.

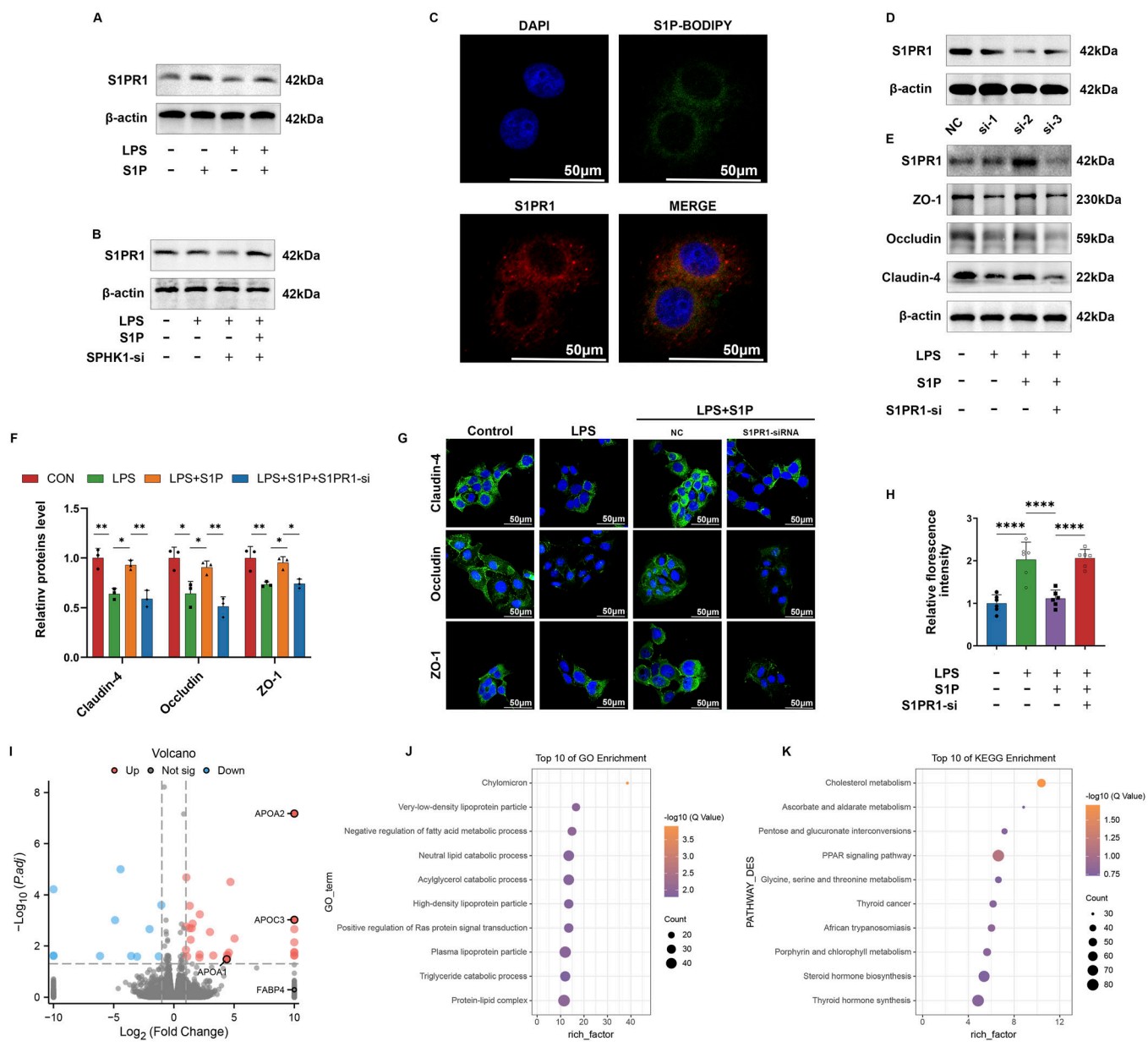

**Figure 5. S1P exerts a protective effect on TJs through activation of S1PR1.**
**(A)** WB analysis of S1PR1 under LPS, S1P addition. **(B)** WB analysis of S1PR1 under LPS, S1P addition after completion of knockdown of SPHK1. **(C)** Fluorescence imaging of S1P with S1PR1 by externally attached fluorescent motif BODIPY. **(D)** Verification of knockdown efficiency of S1PR1-siRNA. **(E)** WB analysis of TJs in cellular proteins with LPS or S1P addition after completion of knockdown of S1PR1. **(F, J)** Semi-quantitative analysis of TJs in (J) using ImageJ software (n = 3). **(G)** IF imaging of TJs under the addition of LPS or S1P after the knockdown of S1PR1. **(H)** Relative fluorescence intensity analysis of epithelial permeabilization under the intervention of LPS, S1P after completion of knockdown of S1PR1 (n = 6). **(I, J, K)** Whole-gene transcriptomics sequencing of LPS and LPS+S1P group cells, mapping differential gene volcano and GO, KEGG enrichment analysis bubble maps. Bars represent mean ± SE, *P < 0.05, **P < 0.01, ***P < 0.001, ****P < 0.0001 determined by one-way ANOVA. Source data are available for this figure.

membranes and found it revealed periplasmic cholesterol accumulation in the LPS-treated group. Addition of S1P notably improved cholesterol distribution, resulting in a more uniform and central arrangement within the intercellular space and along the cell membrane. After PPAR-α knockdown, cholesterol accumulation worsened in the LPS-treated group compared with the control LPS group (Fig 6H).

It is evident that S1P activation of S1PR1 stimulates the PPAR-α pathway, increasing expression of the downstream lipoproteins APOA1, APOA2, and APOC3, which synergistically enhance cholesterol transport and its periplasmic distribution. This mechanism is crucial for S1P's protective effects on TJs (Fig 7). The diagram illustrating this mechanism was drawn by Figdraw.

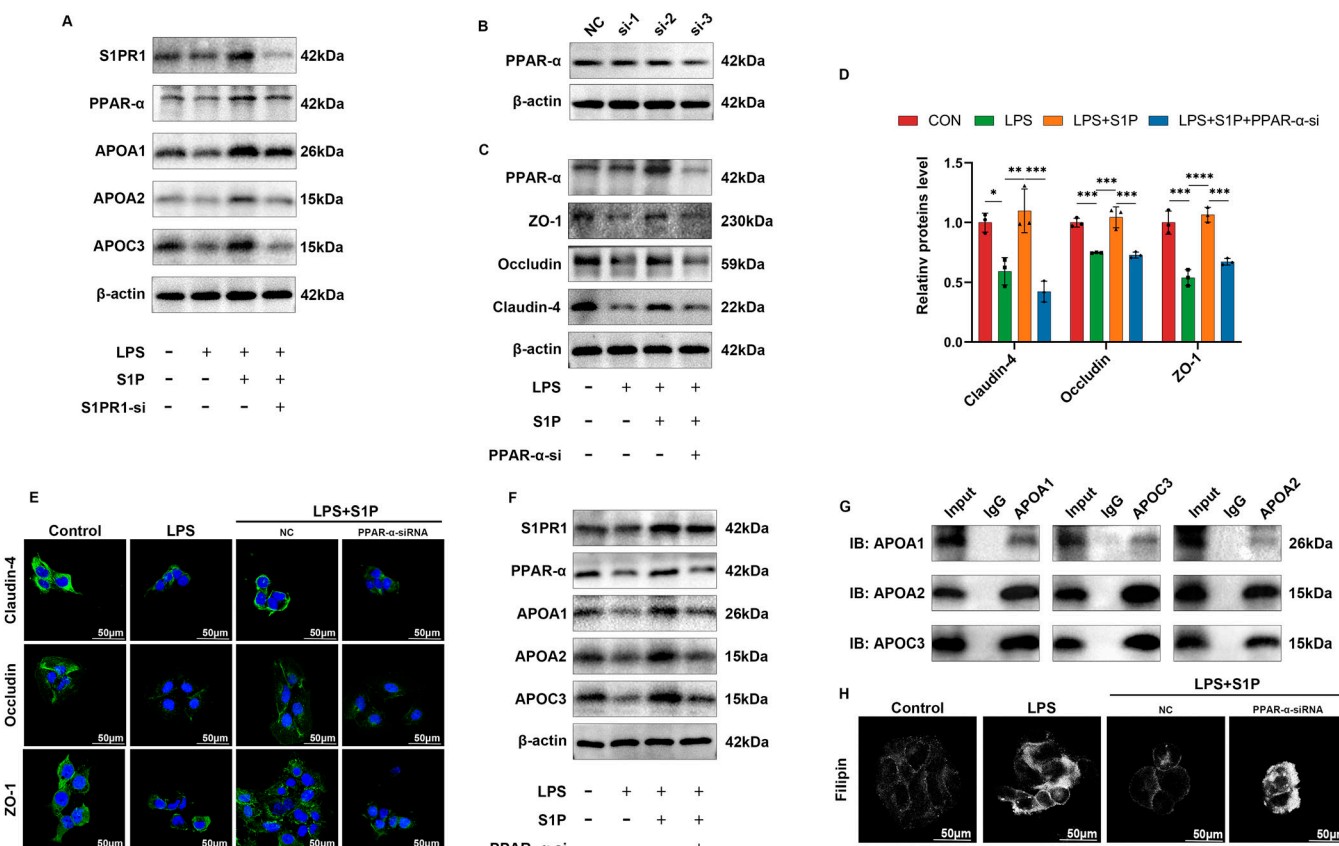

**Figure 6. S1PR1 restores TJs expression by activating the PPAR-α pathway to promote cholesterol transport.**
**(A)** WB analysis of PPAR-α and its downstream pathway proteins, APOA1, APOA2, APOC3, in proteins of cells with LPS or S1P added after completion of knockdown of S1PR1. **(B)** Verification of knockdown efficiency of PPAR-α-siRNA. **(C)** WB analysis of TJs in proteins of cells with the addition of LPS or S1P after completion of knockdown of PPAR-α. **(D, E)** Semi-quantitative analysis of three kinds of TJs in (E) using ImageJ software (n = 3). **(E)** IF imaging of TJs under the addition of LPS or S1P after completion of knockdown of PPAR-α. **(F)** WB analysis of PPAR-α and its downstream pathway proteins APOA1, APOA2, and APOC3 in cellular proteins added with LPS or S1P after completion of knockdown of PPAR-α. **(G)** CO-IP results of PPAR-α downstream lipoproteins. **(H)** Cholesterol staining of Sv-huc-1 under LPS or S1P treatment using Filipin after completion of knockdown of PPAR-α. Bars represent mean ± SE, *P < 0.05, **P < 0.01, ***P < 0.001, ****P < 0.0001 determined by one-way ANOVA. Source data are available for this figure.

# Discussion

The bladder epithelial barrier, a biological structure formed spontaneously by the body, prevents tissue inflammation by blocking the infiltration of urine and its toxic substances (29). Disruption of tight junctions between epithelial cells, a critical component, increases permeability and leads to lower urinary tract symptoms (30). Gene expression profiling revealed significant down-regulation of tight junction proteins, such as ZO1 and occludin, in the bladder epithelial tissues of IC/BPS patients, indicating compromised tight junction function in this condition (31). Manipulation of the bladder epithelial junction protein Claudin2 via gene editing in IC/BPS rat models increased bladder epithelial barrier permeability and damage, accompanied by irritative voiding symptoms and pelvic pain (32). Increasing bladder uroepithelial permeability with protamine sulfate significantly heightened the bladder's mechanical sensitivity to graded dilatation (14). Based on these observations, we hypothesize that altered epithelial permeability plays a crucial role in the development of lower urinary tract symptoms in IC/BPS patients. S1P, a molecule pathologically

elevated in IC/BPS, regulates physiological functions through S1PR activation and is a key signaling molecule in maintaining the barrier function of epithelial and endothelial cells (16, 33, 34). S1P and related signaling pathways maintain barrier integrity by regulating tight junction assembly, cytoskeletal rearrangement, and adhesion patch formation (35, 36). Studies have demonstrated that S1P reduces inflammatory cytokine production and mitigates disruption of the lung epithelial barrier in LPS-induced inflammation models (37). In vascular endothelial cells, the S1P/S1PR1 signaling pathway modulates responses to immune complex deposition, enhances barrier integrity, and reduces inflammatory damage to end organs (38). Although S1P is crucial for maintaining the integrity of both the vascular and intestinal epithelial barriers in vivo, its role in the bladder epithelial barrier remains to be clarified (39). Investigating whether a specific increase in S1P can improve lower urinary tract symptoms by enhancing the bladder epithelial barrier is of significant interest.

In this study, we developed a mouse model displaying lower urinary tract symptoms by inducing uroepithelial disruption with CYP (40). Mice with CYP-induced acute cystitis exhibited increased

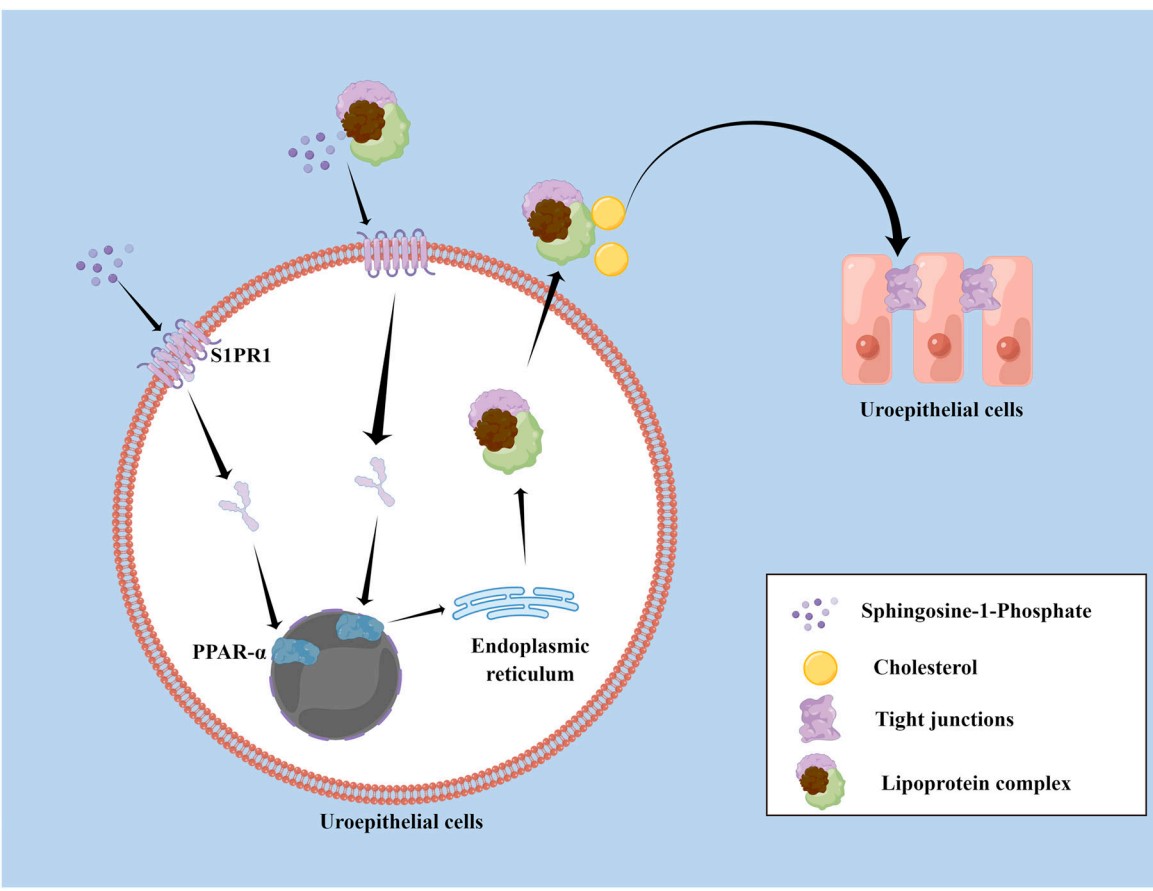

**Figure 7.  Diagram of the mechanism by which S1P/S1PR1 repairs TJs via PPAR-α.**

voiding frequency and decreased urine output, as evidenced by smaller and more numerous urine spots on filter paper. In addition, these mice showed signs of lower abdominal pain, reflected in their pain scores. To administer a higher concentration of the drug, we performed bladder instillations of S1P under anesthesia. IC/BPS-like LUTS in mice were alleviated after S1P bladder instillation. Further experiments demonstrated that S1P not only restored the uroepithelial cell layer and repaired tight junctions (TJs) but also highlighted that SPHK1−/− mice suffered more severe bladder injury from CYP. However, it remains unclear whether S1P's action promotes proliferative repair of the uroepithelial layer or enhances the formation of intercellular tight junctions. To further investigate S1P's effect on tight junctions, we employed an LPS-induced sv-huc-1 barrier injury model. Initially, we created two cellular states: S1P enrichment and depletion, through exogenous addition of S1P and SPHK1 knockdown, respectively. This model provides new evidence that S1P can play a protective role in restoring uroepi-thelial TJ. Given S1P's dual role as both a first and second messenger, we tagged exogenously added S1P molecules with the fluorescent BODIPY group. S1P-BODIPY primarily co-localized with S1PR1 and the cell nucleus. The biological function of S1P was lost after S1PR1 knockdown. These results confirm S1P's function as a first messenger.

S1PR1 is critical for maintaining the vascular endothelial barrier. Congenital knockdown in mice leads to abnormal cardiovascular development and adverse outcomes like miscarriage and stillbirth. Acquired knockdown results in increased vascular leakage in the pulmonary system, exacerbating injury and disease progression (41, 42, 43). Our findings regarding the bladder epithelial barrier were consistent, though the underlying mechanisms remain elusive, prompting us to further complete whole-gene transcriptomic se-quencing. Differential gene expression analysis and pathway mapping revealed significant insights into cholesterol metabolism and the PPAR-α signaling pathway. The knockdown of PPAR-α abolished the biological function of S1P mediated by S1PR1, highlighting PPAR-α as a critical downstream protein. After PPAR-α activation by S1PR1, high-density lipoproteins APOA1, APOA2, and APOC3 were up-regulated, facilitating cholesterol transport in the form of binding. Recent research has shown that cholesterol-rich domains of ZO proteins play a crucial role in the composition of intercellular tight junctions (27). Our studies revealed that PPAR-α knockdown reduced the expression of HDLs APOA1, APOA2, and APOC3; led to extracellular cholesterol accumulation; disrupted periplasmic structure; and hindered tight junction restoration. This indicates that cholesterol's role in forming tight junctions involves not just increased quantity but also its transport and distribution.

This study demonstrates that S1P has a protective role in IC/BPS. Contrary to facilitating the destruction, exogenous addition of S1P actually protects TJs in the inflammatory state of uroepithelial cells. Subsequently, bladder infusion of S1P alleviated lower urinary tract symptoms in mice with cystitis and restored the distribution of TJs both within and between uroepithelial cell layers. S1P functions primarily as a first messenger, where binding of S1P leads to the internalization of the S1PR1 receptor, which activates the PPAR-$\alpha$ signaling pathway. This activation enhances the expression of lipoproteins—APOA1, APOA2, and APOC3—synergistically facilitating cholesterol transport. This activation not only improves the periplasmic distribution of cholesterol but also promotes the recovery of TJs. Therefore, S1P bladder perfusion represents a promising therapeutic strategy for protecting the urinary epithelial barrier and effectively treating IC/BPS.

# Materials and Methods

### Mice

The Animal Center at Nanjing Medical University provided female C57BL/6J mice weighing 18 and 20 g and aged 8–10 wk. SHPK1–/– mice were purchased from Cyagen Biosciences Inc. All mice were housed in individual cages with a 12-h light/dark cycle and unfettered access to food and water at 20–26°C and 40–60% relative humidity. Under isoflurane anesthesia, mice were euthanized by cervical dislocation in this study. All experimental procedures were approved by the Animal Ethical and Welfare Committee of Nanjing Medical University (IACUC-2210019).

### Mouse model of cystitis induced by CYP

Mice were divided into three groups: control, CYP (C0768; Sigma-Aldrich) + vehicle, and CYP+ S1P (B6707; APExBIO) group. To establish a mouse IC model, CYP was injected intraperitoneally at a concentration of 15 mg/ml at a dose of 150 mg/kg body weight. In contrast, saline was injected intraperitoneally into the control group. Mice were euthanized 24 h after administration. The mice within the same group were treated equally.

### Sphingosine 1-phosphate (S1P) bladder insufflation

24 h after the CYP injection, the mice were under anesthesia with 2% isoflurane for ~30 min. After the bladder was emptied by applying mild finger pressure to the lower abdomen, mice were perfused with 50 $\mu$l of S1P or vehicle in a 1-ml syringe with a 24 g indwelling needle jacket and maintained for 30 min. During anesthesia, care was taken to maintain the mice's body temperature and respiratory heart rate.

### VSA

The frequency of urination and the average volume were studied using the VSA (44). A mouse was carefully placed in a metabolic cage (Yuyan) with a piece of filter paper (Whatman No. 3) affixed to the bottom. During the experiment, the mice were provided with a dark, silent environment, standard food, and water and were kept in a room with constant temperature and humidity. At the end of the 2-h observation period, the filter paper was removed from the cage, and the urine stains were imaged using UV light and a transmission illuminator. The Fiji variant of the ImageJ software was used to analyze the images. The area-to-volume standard curve used to calculate urine volumes is provided in supplementary materials.

Considering that the frequency of urination in C57BL/6J mice varies throughout the day, VSA experiments were conducted between 09:00 and 16:00 every day at approximately the same time. The exclusion of suspicious urine stains (feces and mice scratch marks) was performed.

### Measurement of pelvic mechanical sensitivity

The nociceptive response of the pelvis was evaluated 24 h after the CYP injection (45). Before observation, mice were acclimatized on a raised wire lattice (Yuyan) for 1 h. The lower abdomens of mice were sequentially mechanically stimulated with four filaments of 0.008, 0.07, 0.4, and 1.0 g (North Coast, America). The assessment of the implicated nociceptive hypersensitivity was based on the after criteria: 0 points: the experimental mouse did not respond to stimulation; 1 point: the mouse flexed the trunk, punched the limbs, lifted the upper body, and departed the position; 2 points: the mouse jumped. The cumulative score was recorded after each mouse was stimulated 10 times with the same filament.

### Urodynamic measurement

Mice were treated as described above and then anesthetized with an isoflurane (2%) nebulizer, positioned flat on a fixation plate, and the abdominal cavity was opened from the lower abdomen medially. The bladder was progressively exposed to separate it, after which a scalpel needle was inserted and secured into the bladder. The scalp needle was connected to a bi-directional valve connected to a pressure transducer (Taimeng) and microinjection pump (Silugao), and the air was evacuated from the entire line. RT saline was injected into the bladder at a rate of 1 ml/h to induce the repetitive bladder contractions. The maximal systolic bladder pressure and inter-systolic interval were detected and recorded using a multi-channel signal processing system (Taimeng) (46). For the abovementioned factors, we chose 9:00–16:00 for the experiment.

### Histopathological analysis

Fresh bladders were obtained and washed using PBS, fixed in 4% PFA for 24 h, and embedded in paraffin. The tissues were cut into 4-$\mu$m sections, stained using hematoxylin and eosin (H&E), and photographed under a microscope at 40x and 200x magnification.

### Cell culture and treatment in vitro

SV-HUC-1 cells were obtained from Zhong Qiao Xin Zhou Biotechnology (ZQ0345) and cultured in F-12K medium supplemented with 10% FBS and 1% penicillin-streptomycin solution (ZQ-522;

Zhong Qiao Xin Zhou Biotechnology). 5% $CO_2$ concentration and 37°C were provided. We used LPS (L3129; Sigma-Aldrich) with continuous stimulation for 24 h to construct a cellular barrier injury model and exogenously added S1P to simulate the condition of cellular S1P enrichment.

## Western blotting

We separated the bladder epithelial layer under a microscope with a frontal view. Cells in six-well plates were washed twice with PBS and set aside. Proteins were extracted using a RIPA lysis buffer (Beyotime). The protein concentration was measured using BCA (Beyotime), adjusted to a homogenous concentration by adding loading buffer and ddH$_2$O, and stored in a refrigerator at −20°C. Protein aliquots were separated using 30% sodium dodecyl sulfate-polyacrylamide gel electrophoresis (SDS–PAGE) gels and transferred to polyvinylidene fluoride (PVDF) membranes. After 30 min of a block at RT with Rapid Blocking Buffer (TBST-T) powder, the membranes were incubated overnight at 4°C with primary antibodies. After incubation with horseradish peroxidase-conjugated secondary antibodies, antibody-antigen complexes were detected using ECL substrates and visualized with an imaging system.

Primary antibodies were as follows: ZO-1 (rabbit, 1:2,000, 21773-1-AP; Proteintech), occludin (rabbit, 1:1,000, 27260-1-AP; Proteintech), claudin-4 (rabbit, 1:200, 16195-1-AP; Proteintech), E-cadherin (rabbit, 1:5,000, 20874-1-AP; Proteintech), IL-6 (rabbit, 1:500, 21865-1-AP; Proteintech), TNF-$\alpha$ (mouse 1:1,000, 60291-1-Ig; Proteintech), NLRP3 (mouse, 1:1,000, 68102-1-Ig; Proteintech), S1P1 (rabbit, 1:1,000, 55133-1-AP; Proteintech), SPHK1 (rabbit, 1:1,000, 10670-1-AP; Proteintech), uroplakin-IIIa (UPK-IIIa) (mouse, 1:50, sc-166808; SANTA CRUZ), ki67 (rabbit, 1:2,000, 27309-1-AP; Proteintech), cytokeratin5 (krt5) (mouse, 1:200, 66727-1-Ig; Proteintech), cytokeratin20 (krt20) (rabbit, 1:1,000, 17329-1-AP; Proteintech), APOAI (rabbit, 1:1,000, 14427-1-AP; Proteintech), APOAII (mouse, 1:1,000, 66773-1-Ig; Proteintech), APOCIII (rabbit, 1:2,000, ab108205; Abcam), PPAR-$\alpha$ (rabbit, 1:1,000, ab227074; Abcam), and $\beta$-actin (mouse, 1:5,000, 66009-1-Ig; Proteintech). Secondary antibodies were as follows: HRP-conjugated Affinipure Goat Anti-Mouse IgG(H+L) (1:4,000, SA00001-1; Proteintech) and HRP-conjugated Affinipure Goat Anti-Rabbit IgG(H+L) (1:4,000, SA00001-2; Proteintech).

## IHC and immunofluorescence (IF)

Animal tissue sections or cell slides were prepared and blocked with 5% (cell) or 10% BSA (animal) for 30 min, then incubated with primary antibody (see above) at 4°C overnight, followed by incubation with secondary antibody (1:200 dilution) for 1 h at RT and protected from light. IHC used DAB staining followed by hematoxylin re-staining of nuclei, and IF used DAPI (Sigma-Aldrich) re-staining of nuclei followed by autofluorescence quenching. Fluorescence images were photographed by Olympus FV3000.

Secondary antibodies were as follows: CoraLite488-conjugated Goat Anti-Mouse IgG(H+L) (1:200, SA00013-1; Proteintech), CoraLite488-conjugated Goat Anti-Rabbit IgG(H+L) (1:200, SA00013-2; Proteintech), CoraLite594-conjugated Goat Anti-Mouse IgG(H+L) (1:200, SA00013-3; Proteintech), CoraLite594-conjugated Goat Anti-Rabbit IgG(H+L) (1:200, SA00013-4; Proteintech).

## Mouse genotyping

Rinse scissors and forceps with 70% ethanol before the experiment. Cut 0.2–1 cm of mouse tail tip and place it in 100 $\mu$l of pre-prepared digestion solution (96 $\mu$l DNA Extraction Solution + 4 $\mu$l Enzyme Mix); you need to make sure that the mouse tail is completely submerged in the solution, and use scissors to cut the tissues to ensure full lysis. Place samples in a 55°C water bath or PCR machine and incubate for 15 min. Place the samples in a 95°C water bath or PCR machine and incubate for 5 min. Add 100 $\mu$l of Stop Solution to the above samples and vortex to mix. Store at 4°C or perform PCR. Primers were designed according to genotypes, and the primer sequences were as follows: F1: 5'-AAAGAGGTATTGCGGTGTCCCAC-3', R1:5'-AGACTGTGGGTGGTGGGTCTG-3'. Primer set 2: F2: 5'-GGATGTTCTAGTGTGGCTGAGTC-3', R2:5'-GTGAGTGTGATTATCCAGACA-CAACAG-3'. Target gene knockdown positive: 632 bp. Target gene not knocked out negative: 463 bp, (−/−): 632 bp, (+/−): 632 bp, 463 bp, (+/+): 463 bp (Fig S2).

## Epithelial permeability assay

SV-HUC-1 cells were inoculated on polyester membrane filters in 24-well Transwell plates (3413; Corning). A culture medium was added to the upper and lower chambers, and after the cells had reached the desired level of fusion, they were treated with intervention for 24 h. Subsequently, the cells were washed with PBS to remove any residual drug, and 500 $\mu$l of fluorescein 5-iso-thiocyanate (FITC)–dextran (FD-4; Sigma-Aldrich) solution (dissolved in HBSS, 1.0 mg/ml) was added. After incubation at 37°C for 1 h, 50 $\mu$l of HBSS was taken into a 96-well plate protected from light, and the fluorescence intensity was measured at an excitation wavelength of 485 nm and an emission wavelength of 530 nm.

## RNA sequencing

Sv-huc-1 cells with exogenously added LPS and LPS+S1P were subject to RNA sequencing (RNA-seq). Total RNA was extracted using the MJzol Animal RNA Isolation Kit (Majorivd) according to the standard operating procedures provided by the manufacturer and purified using the RNAClean XP Kit (Beckman Coulter) and RNase-Free DNase Set (QIAGEN). The purified total RNA was subjected to mRNA isolation, fragmentation, first-strand cDNA synthesis, second-strand cDNA synthesis, end repair, 3' end plus A, ligation of junctions, enrichment, and other steps to complete the construction of the mRNA sequencing library. The library concentration was detected using a Qubit 2.0 fluorescence quantimeter (Thermo Fisher Scientific), and the library fragment distribution was detected using an Agilent 4200 TapeStation (Agilent Technologies). Sequencing was performed according to the effective concentration of libraries and data output requirements. The sequencing platform was Illumina NovaSeq6000, and the sequencing mode was PE150 (Pair-end 150 bp), followed by the bioinformatics analysis results they provided. The criteria for differentially expressed genes was defined as | log$_2$ fold—change (FC) | > 1.0, after the adjustment $P$-values < .05. R software (version 4.2.2) was used to perform Gene Ontology (GO) and Kyoto Encyclopedia of Genes and Genomes (KEGG) enrichment analysis of differentially expressed genes.

## S1P ELISA assay

We used an ELISA kit (EU2603; Fine Test) to detect S1P levels in mouse serum and in the bladder. Serum and bladder tissues were prepared in advance according to the instructions and placed on ice. We rewarmed the kit to RT and washed the plate twice. We added 50 $\mu$l of standard or sample into the corresponding wells and then immediately added 50 $\mu$l of biotin-labeled antibody. The plate was gently tapped for 1 min to ensure thorough mixing then static-incubated for 45 min at 37°C. After washing the plate three times, 90 $\mu$l TMB substrate solution was added, and the plate was static-incubated for 10–20 min at 37°C. After we observed a gradient-like change in the standard wells, 50 $\mu$l of termination solution was added, and the absorbance was immediately measured at 450 nm.

## Statistical analysis

All values represent mean ± SEM. When the F ratio exceeded the critical value at $\alpha$ = 0.05, post hoc multiple comparisons tests were performed. Data were analyzed using a two-way ANOVA with Tukey's multiple comparisons tests, a repeated-measures two-way ANOVA with Brown-Forsythe test, or Bartlett's test, as indicated by experimental design. All analyses were performed using GraphPad Prism software (GraphPad Prism version 8.4.3 for Windows, GraphPad Software).

# Data Availability

All data needed to evaluate the conclusions in the article are present in the article and the Supplementary Materials. Additional data related to this article may be requested from the authors.

# Supplementary Information

# Acknowledgements

This work was supported by the National Natural Science Foundation of China (82270817, B Shen) and the National Natural Science Foundation of China (82370781, Z Wei). We thank Z Dongsheng of the Nanjing Medical University for synthesizing S1P-BODIPY.

## Author Contributions

J Zhang: conceptualization, resources, data curation, formal analysis, validation, methodology, project administration, and writing—original draft, review, and editing.
Q Ge: conceptualization, data curation, formal analysis, methodology, and writing—review and editing.
T Du: conceptualization, data curation, formal analysis, methodology, and writing—review and editing.
Y Kuang: formal analysis, methodology, and writing—review and editing.
Z Fan: data curation and methodology.
X Jia: formal analysis and methodology.
W Gu: data curation and methodology.
Z Chen: conceptualization, formal analysis, validation, methodology, and writing—review and editing.
Z Wei: conceptualization, funding acquisition, validation, project administration, and writing—review and editing.
B Shen: conceptualization, data curation, funding acquisition, validation, investigation, methodology, and writing—review and editing.

## Conflict of Interest Statement

The authors declare that they have no conflict of interest.

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
