## [Reviewer comments · Life Science Alliance]

Life Science Alliance

SPHK1/S1PR1/PPAR- α axis restores TJs between uroepithelium providing new ideas for IC/BPS treatment

Junjie Zhang, Qingyu Ge, Tianpeng Du, Yuhao Kuang, Zongyao Fan, Xinyi Jia, Wenjin Gu, Zhengsen Chen, Zhongqing Wei, and Baixin Shen

DOI: <https://doi.org/10.26508/lsa.202402957>

Corresponding author(s): Baixin Shen, Second Affiliated Hospital of Nanjing Medical University and Zhongqing Wei, Second Affiliated Hospital of Nanjing Medical University

Review Timeline:

Submission Date:	2024-07-24
Editorial Decision:	2024-09-20
Revision Received:	2024-10-28
Editorial Decision:	2024-11-12
Revision Received:	2024-11-13
Accepted:	2024-11-15

Transaction Report:

September 20, 2024

Re: Life Science Alliance manuscript #LSA-2024-02957-T

Prof. Baixin Shen
The Second Affiliated Hospital of Nanjing Medical University
121 Jiangjiaoyuan
Nanjing 210000
China

Dear Dr. Shen,

Thank you for submitting your manuscript entitled "SPHK1/S1PR1/PPAR- α axis restores TJs between uroepithelium providing new ideas for IC/BPS treatment" to Life Science Alliance. The manuscript was assessed by expert reviewers, whose comments are appended to this letter. We invite you to submit a revised manuscript addressing the Reviewer comments.

Thank you for this interesting contribution to Life Science Alliance. We are looking forward to receiving your revised manuscript.

Sincerely,

B. MANUSCRIPT ORGANIZATION AND FORMATTING:

Reviewer #1 (Comments to the Authors (Required)):

In this manuscript, the authors examine the role of sphingosine-1-phosphate (S1P) signaling in the bladder and its role in the etiology of interstitial cystitis/bladder pain syndrome (IC/BPS). The authors do an excellent job in describing well-designed and controlled experiments that show that S1P signaling through PPAR- α is protective against urothelial cell death, urothelial permeability, as well as pelvic nociception and bladder hyperactivity. The data presented is comprehensive and, for the most part, supports the authors' conclusions. I have only a few comments/concerns.

Major Concerns:

1. Methods:

- a. Why were daylight hours picked for void-spot assay recordings? Rodents are nocturnal, so they will be asleep during daylight hours, greatly reducing their movement and limiting the usefulness of the void spot analysis.
 - b. Isoflurane is not a suitable anesthetic for rodent cystometry, as it causes urinary retention in >75% of animal due to inhibition of the urethral sphincter.
 - c. IHC/IF: please define what a "cell crawl" is.
2. Why was an acute CYP treatment chosen over a chronic? The chronic treatment would be much more clinically relevant.
3. The authors state that CYP destroyed tight junctions, which is evidenced by a decrease in ZO-1, occludin and Claudin 4 expression. However, according to the authors' IHC experiments, CYP appears to destroy the umbrella cell layer completely, so there are longer any cells that express TJs in the urothelium. I think the correct statement would be that CYP causes apoptosis of umbrella cells (something the authors also show) and S1P is protective against that. This is another reason to use the chronic model of CYP treatment, as it results in much less urothelial cell death. Also, what is the difference between the data in Figure 2A, which shows a complete absence of TJ proteins with CYP and the data in Figure 3E, which shows relatively minor decreases in TJ protein expression for WT CYP?
4. The LPS data presented later supports the hypothesis of TJ-protective effects of S1P, as LPS does not result in massive urothelial cell death. However, it is not a model of IC/BPS, but of bacterial cystitis.

Minor Concerns:

1. The English grammar needs to be improved prior to publication. There are many instances of incorrect grammar particularly in the methods and results sections.

Reviewer #2 (Comments to the Authors (Required)):

SPHK1/S1PR1/PPAR- α axis restores TJs between uroepithelium providing new ideas for 1 IC/BPS treatment

1. A short summary of the paper, including description of the advance offered to the field.
2. For each main point of the paper, please indicate if the data are strongly supportive. If not, explicitly state the additional experiments essential to support the claims made and the timeframe that these would require.
3. Lastly, indicate any additional issues you feel should be addressed (text changes, data presentation, statistics etc.)

The above paper by Zhang et al., addresses a very important pathological condition in humans across the world using the routinely used chemical cystitis model using cyclophosphamide (CYP) in an acute treatment paradigm. The authors performed standard end point tests like Void spot assay, measurement of pelvic mechanical sensitivity using von frey filament, etc., along with molecular and pathological analyses to establish the effect of CYP on the urinary bladder and its cells. Owing to its role in enforcing junctional stability and cell proliferation in endothelial systems, the authors pursue an interesting hypothesis that S1P treatment would counteract the damaging effects of CYP on the uroepithelial cells. The alleviating effects of S1P in the model is evidenced using bladder instillation of S1P and using Sphk1 $-/-$ mice model. Later the authors show that PPAR- α downstream of S1P/S1PR1 regulates the barrier phenotypes in vitro, in trying to describe the mechanism of remediation of the uroepithelial damage phenotypes in vivo. Despite the paper trying to drive a new perspective into the treatment of IC/BPS, there are some caveats which the authors need to address to earn the merit of publication of this story.

Major comments:

1. The acute model of cystitis using CYP fares well in terms of the end point measurements. The effect of S1P bladder instillation is even more exciting with the dramatic restoration of junctional proteins observed by western blotting as well as IHC. However, there are many gaps that need to be addressed for the benefit of the field.
 - a. Does S1P instillation change the S1P concentration of the uroepithelial tissue? This can be achieved by using previously published LC-MS/MS methods or by using S1P specific antibodies to stain for the same.
 - b. In the methods section it is mentioned that 50 μ L of S1P was perfused. Fig 1B shows a dose-response experiment using the void spot assay. In the text, the authors must be more explicit about the concentration of S1P perfused in all subsequent experiments.
 - c. If the S1P reaches the inside of the bladder, to understand the timescales of S1P action within the next 24 hours, what is the kinetics of the phenotypic changes that one observes in the tissues? I would recommend checking void spot assays and IHC for ZO-1, Occludin, Claudin-4 or H&E on the bladder tissues like in Fig 1E, for time-points 0, 4, 16 and 24 hours post S1P treatment.
 - d. Since the mechanism proposed by the authors is S1P dependent, I would recommend them to determine expression of S1PR1-5 by qPCR or western blotting from these tissues from uroepithelial cells isolated from the bladder like in Fig 2A.
2. I understand the importance of supplementing a direct metabolite addition experiment with a genetic experiment and in this regard, using Sphk1 $-/-$ is a step in the right direction. However, the S1P effect described in Fig1 and 2 are acute and in the timescale of just 24 hrs as opposed to the developmental timescale of Sphk1 $-/-$ mice which is a full body knockout. Ideally, an inducible KO mice would serve as the best experiment. I would suggest two experiments here:
 - a. Use a heterozygous control (Sphk1 $+/-$) to show the effect of S1P in a dose-dependent manner in CYP-induced cystitis.
 - b. The whole-body KO mice does not shed light on which cell types in the bladder provide the S1P from Sphk1 activity. Moreover, previous publications (PMID: 16314531, PMID: 17292973) have alluded to compensatory activity by Sphk2 in the absence of Sphk1, and therefore, it would be prudent to check S1P levels from the bladder tissues of the Sphk1 $-/-$ and Sphk1 $+/-$ mice as compared to WT mice.
3. The authors mention on two occasions in lines 316 and 338, that the limitations of their in vivo experiments precluded analysis of restoration of intercellular tight junctions. Can the authors elaborate on this? Do they mean that they are limited by the resolution of imaging to distinguish the signal from tight junction proteins from adjacent cells?
4. The authors transition to an in vitro model to underpin the mechanism of S1P-mediated rescue of the phenotypes. However, in order to do so, they pivot to LPS treatment to cause inflammation on SV-huc-1 cells instead of using acrolein directly to keep parity with the in vivo model of CYP treatment, where it is known that acrolein causes uroepithelial damage. PMID: 35492615 shows the use of acrolein-induced urothelial cell damage in in vitro primary porcine Urothelial Cell pointing to the feasibility of in vitro experiments using Acrolein directly. The authors should consider using acrolein instead of/along with LPS treatment.
5. Figure 6C shows internalization of S1PR1 upon S1P treatment. However, it is not discernible by naked eye as to which S1PR1 signal has internalized in the LPS + S1P panel. I would recommend including a panel with No treatment, where the cells are serum starved for atleast 2-4 hours. Following this, treatment with LPS or LPS + S1P can show clearer internalization patterns. This data should be quantified and plotted graphically.
6. The authors mention their previous studies detecting S1P from urine samples of IC/BPS patients. This could be potentially counter-intuitive considering the central hypothesis of this paper being S1P treatment is theorized as a therapeutic intervention. Measurement of urine S1P from CYP-treated mice should provide some clarity on this point.

Minor comments:

1. Please mention statistical test names for each of the analysis in the corresponding legends. Currently, only p values are mentioned in most of the legends.
2. Figure 7 showing the summary claims that Cholesterol transport upon treatment with S1P is enhanced and that directly enhances the barrier. This is not supported by the experiments in the current paper and should be revised. Moreover, the authors should speculate and label the cell type that presents the S1PR1 receptor which in turn relays the signal to uroepithelial cells.
3. In Fig2A and Fig3E, the mice treated with CYP only (lane 2 in both figures) show harrowing differences in the levels of junctional proteins, although in my opinion, they should be very similar due to similarity of the groups. This needs to be addressed by the authors. I would request authors to provide raw files for all western images showing uncropped full blots.
4. Can the authors comment on the role of endothelial cells in the urinary bladder in the progress of CYP induced cystitis? Does S1P travel from the bloodstream to the basal epithelial cells which crosstalk to the intermediate and superficial umbrella cells to maintain the barrier?

Dear Editors and Reviewers:

Thank you for your letter and for the reviewers' comments concerning our manuscript entitled "SPHK1/S1PR1/PPAR- α axis restores TJs between uroepithelium providing new ideas for IC/BPS treatment" (ID: LSA-2024-02957-T). Those comments are all valuable and helpful for revising and improving our paper, as well as the important guiding significance to our research. We have studied the comments carefully and have made corrections, which we hope to meet with approval.

The revisions in the manuscript are in red fonts, and responses to the reviewers' comments are in blue fonts.

Reviewer #1 (Comments to the Authors (Required)):

In this manuscript, the authors examine the role of sphingosine-1-phosphate (S1P) signaling in the bladder and its role in the etiology of interstitial cystitis/bladder pain syndrome (IC/BPS). The authors do an excellent job in describing well-designed and controlled experiments that show that S1P signaling through PPAR- α is protective against urothelial cell death, urothelial permeability, as well as pelvic nociception and bladder hyperactivity. The data presented is comprehensive and, for the most part, supports the authors' conclusions. I have only a few comments/concerns. Thank you very much for your attention and time.

Major Concerns:

1. Methods:

a. Why were daylight hours picked for void-spot assay recordings? Rodents are nocturnal, so they will be asleep during daylight hours, greatly reducing their movement and limiting the usefulness of the void spot analysis.

Response to comment: Thanks for the comments. Since mice are nocturnal, the reviewer's suggestion that mice are more active at night and sleep during the day is quite reasonable. Hill et al. suggested that due to several factors, including experimental conditions, almost all published VSA data were collected during regular laboratory hours and could support the conclusions (PMID: 30156116). Due to laboratory conditions and the 12h light and dark cycle provided by the animal center, we chose to minimize group differences by strictly controlling the time interval of the assay (09:00-16:00), placing the CON group on filter paper at the same time as the treatment group, and providing a dark and quiet environment.

b. Isoflurane is not a suitable anesthetic for rodent cystometry, as it causes urinary retention in >75% of animal due to inhibition of the urethral sphincter.

Response to comment: Thanks for the comments. Urinary retention was indeed observed during our preliminary experiments, where we considered various factors such as concentration and gas flow rate. After a long period of experimental exploration, we adjusted the concentration of isoflurane and the gas flow rate so that the mice retained their autonomic function (the mice would have a contraction response when their toes were pinched) to maintain the normal bladder contraction function. Other single intraperitoneal injections of anesthetics cannot form a stable drug concentration in mice due to the process of absorption and metabolism, which may interfere with the experimental results.

c. IHC/IF: please define what a "cell crawl" is.

Response to comment: We apologize for the mistake. The correct expression should be 'cell slides'.

2. Why was an acute CYP treatment chosen over a chronic? The chronic treatment would be much more clinically relevant.

Response to comment: Thanks for the comments. IC/BPS is a disease with a chronic course and there is still no best animal model that mimics the disease course. As the reviewer stated chronic models seem to be more clinically relevant in terms of disease course, we explored

various models before the start of the study. We found hyperplastic and thickened changes in the urinary epithelium in mice with the CYP-induced chronic inflammation model, which is not consistent with the exfoliation of epithelial tissue seen in HE staining of clinical patients. The current CYP-induced acute cystitis model is widely used in many studies and can provide arguments to support their conclusions (PMID: 36670953, PMID: 35648087, PMID: 28954463). In our opinion the CYP-induced acute cystitis model represents the early stage of reversible changes in the disease and has better clinical implications for early intervention. Therefore, we chose the CYP-induced acute cystitis model, which is more suitable for the purpose of the study.

3. The authors state that CYP destroyed tight junctions, which is evidenced by a decrease in ZO-1, occludin and Claudin 4 expression. However, according to the authors' IHC experiments, CYP appears to destroy the umbrella cell layer completely, so there are longer any cells that express TJs in the urothelium. I think the correct statement would be that CYP causes apoptosis of umbrella cells (something the authors also show) and S1P is protective against that. This is another reason to use the chronic model of CYP treatment, as it results in much less urothelial cell death. Also, what is the difference between the data in Figure 2A, which shows a complete absence of TJ proteins with CYP and the data in Figure 3E, which shows relatively minor decreases in TJ protein expression for WT CYP?

Response to comment: Thanks for the comments. Renal metabolism of CYP produces acrolein, a metabolite of uroepithelial toxicity, which is more likely to damage the urinary epithelium directly than to cause epithelial cell shedding after apoptosis. Just as the urinary epithelium sheds itself in response to E. coli infection to carry away the flora (PMID: 28861072, PMID: 22874553, PMID: 38101412). Apoptosis is more likely to be the result of inflammation following epithelial barrier damage.

The difference in protein expression can be seen in the data in Figure 2A and Figure 3E, which may be due to several reasons: 1. Due to the preciousness of SPHK1 ^{-/-} mice, we did not strip the epithelium individually to extract the proteins, but rather extracted the whole bladder proteins. During the WB process we would go to increase the volume of up-sampling to validate the TJs protein. 2. The bladder of the CYP group mice in Figure 2A underwent bladder perfusion with 50 μ L of 0.3 M NaOH solution (solvent for S1P) under anesthesia, which may further stimulate the shedding of the urinary epithelium in the injured state.

4. The LPS data presented later supports the hypothesis of TJ-protective effects of S1P, as LPS does not result in massive urothelial cell death. However, it is not a model of IC/BPS, but of bacterial cystitis.

Response to comment: Thanks for the comments. PS is widely used to induce inflammatory response and is a reliable modeling method. LPS (+thyrosin) is able to induce cystitis manifestations in mice, which is one of the modeling methods in the IC/BPS animal model (PMID: 24561047, PMID: 28954463).

The metabolite of CYP, acrolein, has been used to induce inflammation in primary cultured porcine bladder urothelial cells (PMID: 35492615), but there are fewer studies supporting this conclusion. We therefore chose the more commonly used LPS for cellular inflammation modeling. In our in-vitro experiments we focused more on the alteration of cellular tight junctions in the inflammatory state and confirmed the protective role played by S1P in it as well as the mechanism of action.

Minor Concerns:

1. The English grammar needs to be improved prior to publication. There are many instances of incorrect grammar particularly in the methods and results sections.

Response to comment: Thanks for the comments, we've carefully revised the grammar in the methods and results sections.

Reviewer #2 (Comments to the Authors (Required)):

SPHK1/S1PR1/PPAR- α axis restores TJs between uroepithelium providing new ideas for 1 IC/BPS treatment

1. A short summary of the paper, including description of the advance offered to the field.
2. For each main point of the paper, please indicate if the data are strongly supportive. If not, explicitly state the additional experiments essential to support the claims made and the timeframe that these would require.
3. Lastly, indicate any additional issues you feel should be addressed (text changes, data presentation, statistics etc.)

The above paper by Zhang et al., addresses a very important pathological condition in humans across the world using the routinely used chemical cystitis model using cyclophosphamide (CYP) in an acute treatment paradigm. The authors performed standard end point tests like Void spot assay, measurement of pelvic mechanical sensitivity using von frey filament, etc., along with molecular and pathological analyses to establish the effect of CYP on the urinary bladder and its cells. Owing to its role in enforcing junctional stability and cell proliferation in endothelial systems, the authors pursue an interesting hypothesis that S1P treatment would counteract the damaging effects of CYP on the uroepithelial cells. The alleviating effects of S1P in the model is evidenced using bladder instillation of S1P and using Sphk1 ^{-/-} mice model. Later the authors show that PPAR- α downstream of S1P/S1PR1 regulates the barrier phenotypes in vitro, in trying to describe the mechanism of remediation of the uroepithelial damage phenotypes in vivo. Despite the paper trying to drive a new perspective into the treatment of IC/BPS, there are some caveats which the authors need to address to earn the merit of publication of this story.

Major comments:

1. The acute model of cystitis using CYP fares well in terms of the end point measurements. The effect of S1P bladder instillation is even more exciting with the dramatic restoration of junctional proteins observed by western blotting as well as IHC. However, there are many gaps that need to be addressed for the benefit of the field.

a. Does S1P instillation change the S1P concentration of the uroepithelial tissue? This can be achieved by using previously published LC-MS/MS methods or by using S1P specific antibodies to stain for the same.

Response to comment: Thanks for the comments. There are more confounding factors to measure the concentration of S1P after bladder perfusion of synthesized S1P small molecules, so we did not choose this method. According to the reviewer's suggestion we used S1P-BODIPY bladder perfusion followed by frozen sections. KRT20 was used as a uroepithelial marker to localize S1P-BODIPY and to clarify that the site of action of exogenously supplemented S1P was in the uroepithelial layer, and the results were as follows.

b. In the methods section it is mentioned that 50 µL of S1P was perfused. Fig 1B shows a dose-response experiment using the void spot assay. In the text, the authors must be more explicit about the concentration of S1P perfused in all subsequent experiments.

Response to comment: Thanks for the comments. We have indicated in the text that the subsequent S1P perfusion concentration was 400µM. **‘The 400 µM perfusion concentration was chosen for subsequent experiments.’**

c. If the S1P reaches the inside of the bladder, to understand the timescales of S1P action

within the next 24 hours, what is the kinetics of the phenotypic changes that one observes in the tissues? I would recommend checking void spot assays and IHC for ZO-1, Occludin, Claudin-4 or H&E on the bladder tissues like in Fig 1E, for time-points 0, 4, 16 and 24 hours post SIP treatment.

Response to comment: Thanks for the comments. Since CYP-induced acute cystitis is a reversible change, mice themselves respond to the injury and repair the uroepithelium. In our preliminary experiments, we found that the bladder epithelium of mice was almost completely repaired or even covered with thicker uroepithelial cells 48h after CYP treatment, which would interfere with our experimental observation index. After our preliminary exploration, 4h of SIP perfusion is the best observation point for the degree of tissue repair, which can avoid overlapping with the self-repairing effect in mice.

As answering Major Concerns 2 of reviewer 1, our study focuses on early intervention in the reversible stage of the disease. SIP perfusion for 4h mainly plays the role of early intervention, just as early bladder perfusion drugs in clinical IC/BPS patients. The valuable suggestions will be practiced during our subsequent validation of other models.

d. Since the mechanism proposed by the authors is SIP dependent, I would recommend them to determine expression of S1PR1-5 by qPCR or western blotting from these tissues from uroepithelial cells isolated from the bladder like in Fig 2A.

Response to comment: Thanks for the comments. As suggested by the reviewers, we verified the expression of S1PR1-5 after bladder infusion of SIP in mice. S1PR1 expression was elevated and S1PR2-5 was not significantly altered, suggesting that bladder infusion of SIP exerts a biological function through activation of S1PR1. The results were supplemented into Figure S2B.

‘By verifying the protein expression of S1PR1-5 in the bladder of mice in the SIP bladder perfusion group, we found that S1PR1 expression was elevated and S1PR2-5 was not significantly altered (Figure S2B).’

2. I understand the importance of supplementing a direct metabolite addition experiment with a genetic experiment and in this regard, using Sphk1 $-/-$ is a step in the right direction. However, the SIP effect described in Fig 1 and 2 are acute and in the timescale of just 24 hrs as opposed to the developmental timescale of Sphk1 $-/-$ mice which is a full body knockout. Ideally, an inducible KO mice would serve as the best experiment. I would suggest two experiments here:

a. Use a heterozygous control (Sphk1 $+/-$) to show the effect of SIP in a dose-dependent

manner in CYP-induced cystitis.

Response to comment: Thanks for the comments. As stated by the reviewer, SPHK1, as a kinase that generates S1P, has a correlation between its high and low expression and S1P content. We completed the validation of the heterozygous control group during our experiments, the results of which are presented below. Behavioral experiments including voiding imprints as well as urodynamic results showed that the alterations that occurred in SPHK1^{+/-} mice induced by CYP were not significantly different from those in WT CYP mice (A-B). We further completed WB as well as HE staining experiments, and the expression of TJs as well as uroepithelial damage in SPHK1^{+/-} CYP mice also did not show significant differences from WT CYP mice (C-D). This may be because the SPHK1^{+/-} CYP mice still had higher levels of S1P exerting a protective effect.

b. The whole-body KO mice does not shed light on which cell types in the bladder provide the S1P from Sphk1 activity. Moreover, previous publications (PMID: 16314531, PMID: 17292973) have alluded to compensatory activity by Sphk2 in the absence of Sphk1, and therefore, it would be prudent to check S1P levels from the bladder tissues of the Sphk1^{-/-} and Sphk1^{+/-} mice as compared to WT mice.

Response to comment: Thanks for the comments. Our preliminary literature reading found that S1P circulating levels are reduced by 65% in mice after SPHK1^{-/-}, and tissue S1P levels are not reduced (PMID: 15459201). SPHK2^{-/-} mice have normal or elevated levels of S1P circulating (PMID: 16223773, PMID: 22747486), whereas SPHK1-SPHK2 KO mice develop vascular and neural defects (PMID: 16314531). We measured S1P levels in serum as well as bladder of SPHK1^{-/-}, SPHK1^{+/-}, and WT mice using an S1P Elisa Kit as suggested. Serum S1P levels were decreased in SPHK1^{+/-} as well as SPHK1^{-/-} mice and were lower in SPHK1^{-/-} mice than in SPHK1^{+/-} mice (Figure 3A). Bladder S1P levels were elevated in the CYP-induced acute cystitis model (Figure 3B). Bladder S1P levels were significantly decreased in SPHK1^{+/-} and SPHK1^{-/-} mice compared to WT mice, and the decrease was even more pronounced in SPHK1^{-/-} mice.

Methods:

2.14 S1P Elisa Assay

We used an Elisa kit (EU2603, Fine Test, Wuhan) to detect S1P levels in mouse serum as well as in the bladder. Serum and bladder tissues were prepared in advance according to the instructions and placed on ice. We Rewarmed the kit to room temperature and washed the plate twice. We added 50 μl of standard or sample into the corresponding wells, and then immediately added 50 μl of biotin-labeled antibody. The plate was gently tapped for 1min to

ensure thorough mixing then static incubated for 45 minutes at 37°C. After washing the plate three times, 90ul TMB substrate solution were added and the plate was static incubated for 10-20 minutes at 37°C. After we observed a gradient-like change in the standard wells, 50 μl of termination solution was added and the absorbance was immediately measured at 450 nm.'

Results:

'Serum S1P levels were significantly reduced in SHPK1^{+/-} as well as SHPK1^{-/-} mice (Figure 3A). Bladder tissue S1P levels were elevated in CYP-induced cystitis mice, whereas knockdown of SPHK1 reduced S1P levels (Figure 3B).'

3. The authors mention on two occasions in lines 316 and 338, that the limitations of their in vivo experiments precluded analysis of restoration of intercellular tight junctions. Can the authors elaborate on this? Do they mean that they are limited by the resolution of imaging to distinguish the signal from tight junction proteins from adjacent cells?

Response to comment: Thanks for the comments. As stated in the article S1P has a variety of biological effects including promotion of cell proliferation and restoration of intercellular tight junctions. In the in vivo experiments mice experienced significant restoration of the uroepithelium after S1P bladder perfusion as well, which also brings about elevated expression of TJs and fails to focus on the specific pathways of action of S1P. Therefore, we

cultured the uroepithelial cells alone in the in vitro experiments to control the consistency of cell number by means of cell counting and to observe the expression of tight junctions.

4. The authors transition to an in vitro model to underpin the mechanism of S1P-mediated rescue of the phenotypes. However, in order to do so, they pivot to LPS treatment to cause inflammation on SV-huc-1 cells instead of using acrolein directly to keep parity with the in vivo model of CYP treatment, where it is known that acrolein causes uroepithelial damage. PMID: 35492615 shows the use of acrolein-induced urothelial cell damage in in vitro primary porcine Urothelial Cell pointing to the feasibility of in vitro experiments using Acrolein directly. The authors should consider using acrolein instead of/along with LPS treatment. Response to comment: Thanks for the comments. The possible different pathophysiological mechanisms brought about by LPS were an important point of consideration for us during the experimental design. Compared to the widely used LPS, acrolein-induced uroepithelial inflammation is currently less studied to support this conclusion, so we chose the more commonly used LPS for cellular inflammation modelling. In our in vitro experiments we focused more on the alteration of cellular tight junctions in the inflammatory state, and LPS was able to play a role in disrupting the inter-epithelial tight junctions, which satisfied our experimental needs. We confirmed the protective role played by S1P in this as well as the mechanism of action in subsequent experiments. The treatment of acrolein will be used in subsequent experiments, and the reviewers' suggestions are greatly appreciated.

5. Figure 6C shows internalization of S1PR1 upon S1P treatment. However, it is not discernible by naked eye as to which S1PR1 signal has internalized in the LPS + S1P panel. I would recommend including a panel with No treatment, where the cells are serum starved for at least 2-4 hours. Following this, treatment with LPS or LPS + S1P can show clearer internalization patterns. This data should be quantified and plotted graphically.

Response to comment: Thanks for the comments. Due to the outdated version of the referral article, we have removed the CO-IP results and immunofluorescence results of the original Figure 6B-C in the new article and Figure. Considering that the CO-IP results may have non-specific binding and that immunofluorescence does not show the S1PR1 membrane-to-nucleus translocation process very well, we have removed this section. We will follow the reviewer's suggestion in our subsequent study of membrane protein signaling internalization.

6. The authors mention their previous studies detecting S1P from urine samples of IC/BPS patients. This could be potentially counter-intuitive considering the central hypothesis of this paper being S1P treatment is theorized as a therapeutic intervention. Measurement of urine S1P from CYP-treated mice should provide some clarity on this point.

Response to comment: Thanks for the comments. Starting from the urine sequencing results of clinical patients and then introducing mice was the key point of our sequence study. The starting point of mice treated with CYP is the destruction of epithelial function by acrolein, a metabolite of CYP, and the ability of S1P to play a protective role during epithelial injury was the focus of our study. The pathophysiological mechanisms of the mice in the CYP group differed from those of the patients with IC/BPS, and the amount of S1P in their urine was a point of unattended attention for us. We attempted to collect urine in 24-h metabolic cages, but the mice urinated little and frequently resulting in difficulties in collection, and thus failed to complete this experiment.

Minor comments:

1. Please mention statistical test names for each of the analysis in the corresponding legends. Currently, only p values are mentioned in most of the legends.

Response to comment: Thanks for the comments. We have added the name of the statistical test to the legend.

2. Figure 7 showing the summary claims that Cholesterol transport upon treatment with S1P is enhanced and that directly enhances the barrier. This is not supported by the experiments in the current paper and should be revised. Moreover, the authors should speculate and label the

cell type that presents the S1PR1 receptor which in turn relays the signal to uroepithelial cells. Response to comment: Thanks for the comments. Figure 5I-K and Figure 6A show that the expression of apolipoproteins: APOA1, APOA2, and APOC3 are significantly elevated after S1P treatment, and they can function as cholesterol transporters. By knocking down their upstream signaling protein PPAR- α we found that their expression was reduced and they lost their function of protecting tight junctions. Cholesterol protects tight junctions (PMID: 36791108, PMID: 29720382), but APOA1, APOA2, and APOC3 are not cholesterol-producing proteins, so we verified their effects on cholesterol distribution by using Filipin staining. Figure 6H suggests that the distribution of cholesterol after the addition of S1P was uniform and complete in the LPS group as well as the PPS group. was homogeneous and complete, and cholesterol was significantly accumulated in the LPS group as well as the PPAR-siRNA group. Since we used bladder perfusion, S1P directly contacted the uroepithelial cells and activated S1PR1 to activate the downstream pathway, and we have labeled the cell types in Figure 7.

3. In Fig2A and Fig3E, the mice treated with CYP only (lane 2 in both figures) show harrowing differences in the levels of junctional proteins, although in my opinion, they should be very similar due to similarity of the groups. This needs to be addressed by the authors. I would request authors to provide raw files for all western images showing uncropped full blots.

Response to comment: Thanks for the comments. The difference in protein expression can be seen in the data in Figure 2A and Figure 3E, which may be due to several reasons: 1. Due to the preciousness of SPHK1 $-/-$ mice, we did not strip the epithelium individually to extract the proteins, but rather extracted the whole bladder proteins. During the WB process we would go to increase the volume of up-sampling to validate the TJs protein.2, The bladder of the CYP group mice in Figure 2A underwent bladder perfusion with 50 μ L of 0.3 M NaOH solution (solvent for S1P) under anesthesia, which may further stimulate the shedding of the urinary epithelium in the injured state. All remaining raw WB strips have been submitted.

4. Can the authors comment on the role of endothelial cells in the urinary bladder in the progress of CYP induced cystitis? Does S1P travel from the bloodstream to the basal epithelial cells which crosstalk to the intermediate and superficial umbrella cells to maintain the barrier?

Response to comment: Thanks for the comments. We are very sorry that our experimental approach may have caused misunderstanding by the reviewers. We used a bladder perfusion approach to directly supplement S1P, which is a biological function through direct binding of S1P to the S1PR1 receptor in the urinary epithelium. This process activates the downstream PPAR- α pathway and promotes the restoration of intercellular tight junctions, thereby protecting epithelial barrier function.

Bladder endothelial cells may play a role in the secretion of S1P during CYP-induced cystitis, and S1P is able to be transported to the epithelium via apolipoproteins, activating S1PR1 to maintain the barrier. Bladder endothelial cell S1PR1 may also be activated during CYP-induced inflammatory state, enhancing tight junctions between endothelial cells, reducing exudation, and alleviating tissue edema. Our present study takes into account the differences in blood concentration due to hemodynamics and other factors for tail vein injection, and the in vivo level stays at the therapeutic approach of bladder instillation of S1P. The role played by bladder endothelial cells in the progression of CYP-induced cystitis, as suggested by the reviewers, will be our future research direction.

November 12, 2024

RE: Life Science Alliance Manuscript #LSA-2024-02957-TR

Prof. Baixin Shen
Second Affiliated Hospital of Nanjing Medical University
121 Jiangjiaoyuan
Nanjing 210000
China

Dear Dr. Shen,

Thank you for submitting your revised manuscript entitled "SPHK1/S1PR1/PPAR- α axis restores TJs between uroepithelium providing new ideas for IC/BPS treatment". We would be happy to publish your paper in Life Science Alliance pending final revisions necessary to meet our formatting guidelines.

- please be sure that the authorship listing and order is correct
- please add ORCID ID for the secondary corresponding author--they should have received instructions on how to do so
- please add a Summary Blurb/Alternate Abstract and a Category to our system
- please add the Twitter handle of your host institute/organization as well as your own or/and one of the authors in our system
- please be sure that the authorship listing and order are correct
- please consult our manuscript preparation guidelines <https://www.life-science-alliance.org/manuscript-prep> and make sure your manuscript sections are in the correct order
- please add a callout for Figure 1H to your main manuscript text
- you may want to consider uploading Figure 7 as a Graphical Abstract rather than as a figure, but this is up to you

A. FINAL FILES:

B. MANUSCRIPT ORGANIZATION AND FORMATTING:

Sincerely,

Reviewer #2 (Comments to the Authors (Required)):

The response to my comments have been addressed. The authors have also taken care to alter or incorporate texts accurately supporting the conclusions made from the experiments. I would favorably recommend to accept the manuscript.

November 15, 2024

RE: Life Science Alliance Manuscript #LSA-2024-02957-TRR

Prof. Baixin Shen
Second Affiliated Hospital of Nanjing Medical University
121 Jiangjiaoyuan
Nanjing 210000
China

Dear Dr. Shen,

Thank you for submitting your Research Article entitled "SPHK1/S1PR1/PPAR- α axis restores TJs between uroepithelium providing new ideas for IC/BPS treatment". It is a pleasure to let you know that your manuscript is now accepted for publication in Life Science Alliance. Congratulations on this interesting work.

DISTRIBUTION OF MATERIALS:

Again, congratulations on a very nice paper. I hope you found the review process to be constructive and are pleased with how the manuscript was handled editorially. We look forward to future exciting submissions from your lab.

Sincerely,
